# Multi-Temporal Data Fusion in MS and SAR Images Using the Dynamic Time Warping Method for Paddy Rice Classification

**Tsu Chiang Lei [1], Shiuan Wan [2,*], You Cheng Wu [1], Hsin-Ping Wang [3] and Chia-Wen Hsieh [3]**

[1] Department of Urban Planning and Spatial Information, Feng Chia University, Taichung 40724, Taiwan; tclei@fcu.edu.tw (T.C.L.); steven159357@yahoo.com.tw (Y.C.W.)
[2] Department of Information Technology, Ling Tung University, Taichung 40851, Taiwan
[3] Construction and Disaster Prevention Research Center, Feng Chia University, Taichung 40724, Taiwan; t04111@mail.fcu.edu.tw (H.-P.W.), cwhsieh@mail.fcu.edu.tw (C.-W.H.)
[*] Correspondence: shiuan123@teamail.ltu.edu.tw

**Abstract:** This study employed a data fusion method to extract the high-similarity time series feature index of a dataset through the integration of MS (Multi-Spectrum) and SAR (Synthetic Aperture Radar) images. The farmlands are divided into small pieces that consider the different behaviors of farmers for their planting contents in Taiwan. Hence, the conventional image classification process cannot produce good outcomes. The crop phenological information will be a core factor to multi-period image data. Accordingly, the study intends to resolve the previous problem by using three different SPOT6 satellite images and nine Sentinel-1A synthetic aperture radar images, which were used to calculate features such as texture and indicator information, in 2019. Considering that a Dynamic Time Warping (DTW) index (i) can integrate different image data sources, (ii) can integrate data of different lengths, and (iii) can generate information with time characteristics, this type of index can resolve certain classification problems with long-term crop classification and monitoring. More specifically, this study used the time series data analysis of DTW to produce "multi-scale time series feature similarity indicators". We used three approaches (Support Vector Machine, Neural Network, and Decision Tree) to classify paddy patches into two groups: (a) the first group did not apply a DTW index, and (b) the second group extracted conflict predicted data from (a) to apply a DTW index. The outcomes from the second group performed better than the first group in regard to overall accuracy (OA) and kappa. Among those classifiers, the Neural Network approach had the largest improvement of OA and kappa from 89.51, 0.66 to 92.63, 0.74, respectively. The rest of the two classifiers also showed progress. The best performance of classification results was obtained from the Decision Tree of 94.71, 0.81. Observing the outcomes, the interference effects of the image were resolved successfully by various image problems using the spectral image and radar image for paddy rice classification. The overall accuracy and kappa showed improvement, and the maximum kappa was enhanced by about 8%. The classification performance was improved by considering the DTW index.

**Keywords:** remote sensing; time series; data fusion; feature extraction; dynamic time warping; crop phenological information





## 1. Introduction

Paddy rice takes up the largest crop area and has great significance for the global economy, society, and culture. Presently, farmland surveys in various countries are mainly manual surveys, which are time-consuming, labor-intensive, and extremely inefficient. It is hard to conduct a large-scale survey in a short time. However, with the advancement and development of satellite remote sensing detection technology in recent years, farmland monitoring methods have become well-accepted. The use of satellite image data as a monitoring tool along with the use of the machine learning approach has become a major

solution for land cover measurements. This includes supervised learning and unsupervised learning in machine learning. This greatly reduces the manpower and material resources required for agricultural monitoring and management [1,2].

Many types of research have been dedicated to investigating satellite optical data to carry out the target GIS map for delineation of paddy field areas by image classification through the pixel-based method. This includes Maximum Likelihood, Neural Network, Decision Tree, Support Vector Machine, K-means, ISODAT, etc. Different classification methods can be applied by using by the material of Landsat TM and ETM+ series, SPOT series, MODIS, Sentinel-2 and 3, RADARSAT series, ERS-1 and ERS-2, ENVISAT/ASAR, IRS, AVHRR, Sentinel-1A, Aerial-Photo, UAV, etc. In addition, related research is focused on how to increase the accuracy of paddy rice fields. For instance, a series of Vegetation Indicators (VI) and Texture Indicators (TI) may become the proper material for interpretation results and classification accuracy, which can be reinforced. The VI indicators include the Ratio Vegetation Index (RVI), NDVI, Soil-adjusted Vegetation Index (SAVI), etc. The texture indicators include the Gray Level Co-Occurrence Matrix (GLCM), Fractal dimension, Semi-Variogram, etc. Hence, the benefits of statistical analysis and machine learning can be greatly improved. In addition to pixel-based classification, the Region-based Object Classification (ROC) is also well-accepted. There are two main steps for ROC: (1) image segmentation and (2) image classification. The method also renders an effective performance in classification. On the other hand, some scholars have imported Synthetic Aperture Radar (SAR) with multiple time-series data to detect the area of rice fields, which has attracted new attention in recent years. The SAR data are not affected by sunlight. Accordingly, there are many kinds of research that perform Image Fusion (IF) processing between MS and SAR as well [3,4]. Unfortunately, traditional image fusion methods seem to be unable to solve the large differences between the two images, and it is not easy to overcome some practical limitations. For example, the images after fusion are prone to produce unexpected noise [5,6]. The errors of data fusion by multi-period sequence images are often accumulated into classification progress. It generally results in unsatisfactory classification outcomes. That is, the image fusion can not obtain the crop phenological information in detail, which is important for image classification [7]. Rice patches may be affected by mixed crops of land-use on a single patch, different planting seasons, and different varieties, which are governed by different farmer behaviors. In addition, using a single image may result in the interference of cloud and fog effects, which can obstruct the classification results. Furthermore, it destroys the structure of the landscape for considering a single period of texture/vegetation indicator through image fusion.

Therefore, this research does not aim at using the IF methods but uses the concept of the DF method at the starting point. The DF method requires a set of integrated calculations, rather than a method of evaluating data through a single model [8,9]. More specifically, this is a hybrid model to employ different data sources or analysis methods. Related research shows that most of the past studies focused on the fusion of multiple methods [9]. In this study, the data fusion process extracts the variation of features based on various periods with the following considerations. First, the length of time-series changes of different data sources is different. Second, the different properties of data sources vary greatly. Whereas the quality of the original spectral bands must be effectively converted into rational indicators or textures, the different resource data have different resolutions and formats. Third, the method for changing characteristics of patches of different properties vary at different times. It has to effectively extract this relevant information for our research. Hence, DTW is applied in this study for multi-period images for data fusion [10]. It was successfully applied to rice area survey [11,12], landscape changes [10,13], forest type classification [14], farmland mapping [15–20], crop phenological period of factor analysis [21], crop yield estimation [22], etc. This research plans to extract the phenological information in the fragmented landscape from the image through the DTW method to achieve the purpose of rapid mapping, stability, and high accuracy when making rice farmland thematic maps.

Consequently, this study uses the Dynamic Time Warping (DTW) method to compare the similarity results of MS and SAR image datasets by vegetation index and texture index characteristics, respectively. The similarity index results can show the relations in the change of each different land-use of patches. That is, the DTW is examined by the DF method, which contains numerical responses (spectrum, indicators, and textures) to detect similar features in different time series. These relations can improve the classification results efficiently. Hence, three approaches (Support Vector Machine, SVM; Neural Network, NN; and Decision Tree, DT) are used to classify the paddy patches with two groups: (a) without applying the DTW index and (b) considering the DTW index. We adopt the most common classifiers such as SVM, NN, and DT. The goal is not to compare the performance of those three classifiers and determine which one is the best. The key point is to find a possible solution to rationally integrate them with consideration of the function of DTW employing two kinds of image data (optical and radar). The paper contains four steps: Step 1, the first stage of the accuracy of consistency classification; Step 2, the discussion of the accuracy of inconsistency classification; Step 3, examples of multi-scale features and description of integration results; Step 4, the overall accuracy of the hybrid classification.

## 2. Research Materials and Design

### 2.1. Research Materials

2.1.1. Research Site

The study was located at Yunlin County, which is in the Jianan Plain of western Taiwan. It is a major agricultural county for Taiwan. The annual grain output is noteworthy. It mainly produces rice, head, vegetables, peanuts, sweets, and other grains. The coordinates of longitude and latitude are 27′00″120 E, 48′00″23 N. The soil is rich in organic nitrogen, phosphorus, potassium, and other elements, which makes the land abundant in agricultural production. The soil is fertile, and the climate is suitable. Irrigation and rainfall are abundant. Therefore, this study selected Yunlin Xiluo, as in Figure 1. Figure 2a is a map of farmland patches in this area. The total area is about 5016.21 ha with 53,212 patches. Figure 2b is the ground truth data of the study area taken in the first half of 2019 by the Agriculture and Food Agency. Since the main axis of this research is to classify and interpret rice fields, the classification of ground truth data is only divided into paddy rice and non-paddy rice, as in Table 1.

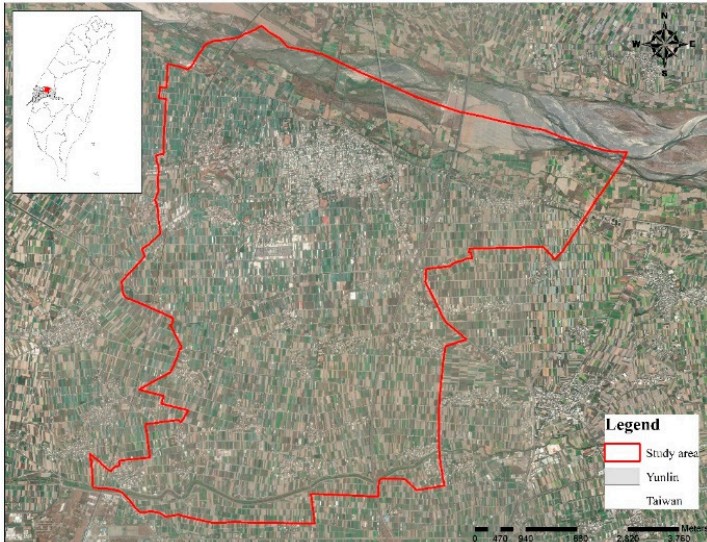

**Figure 1.** Study area.

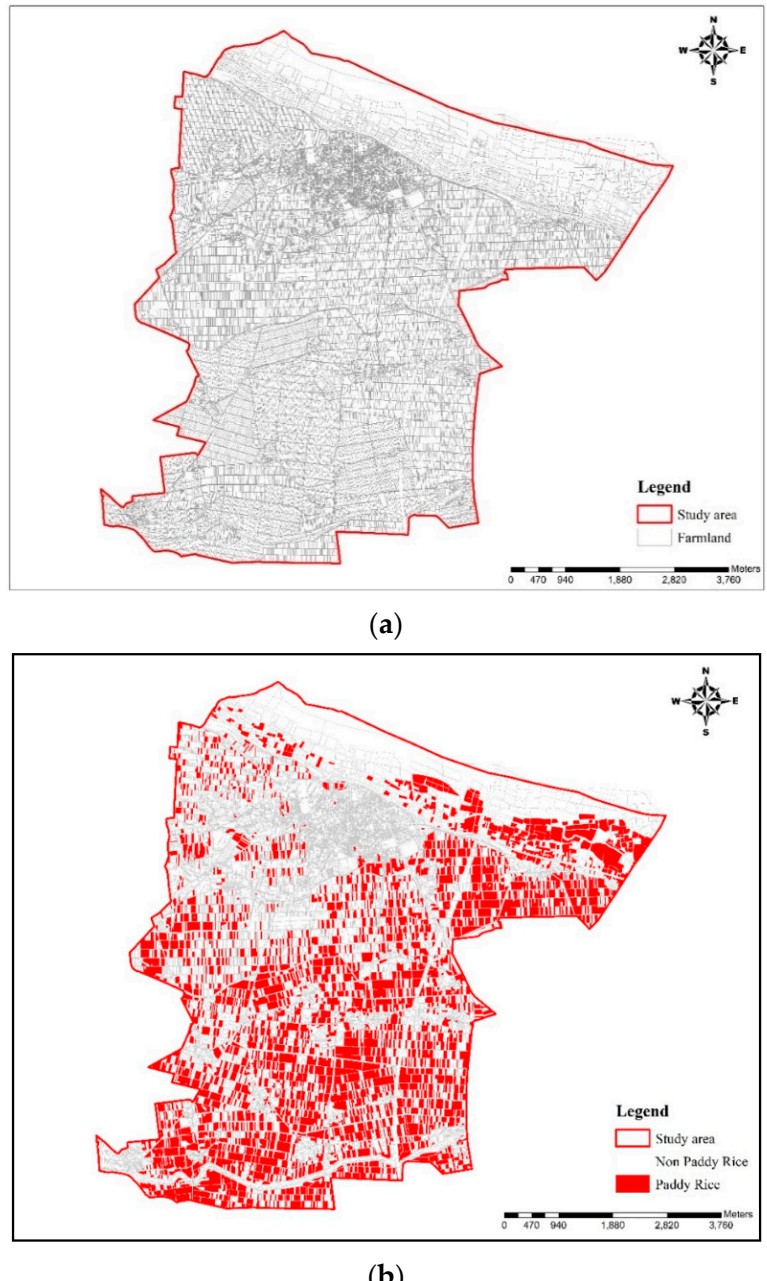

(**a**)

(**b**)

**Figure 2.** The patches are taken from the photo in 2019. (**a**) Farmland patches in study area. (**b**) The ground truth data from the Agriculture and Food Agency.

**Table 1.** The ground truth data of Xiluo in 2019.

| Categories | Number of Patches | Area (ha) |
|---|---|---|
| Paddy Rice | 7699 | 1634.96 |
| Non-Paddy Rice | 45,513 | 3379.83 |
| Total | 53,212 | 5014.80 |

2.1.2. Research Data

SPOT 6 Images

SPOT images basically have four bands, which are multi-spectral images of B, G, R, and IR, with a spatial resolution of 6 m. This study selected SPOT-6 images on 23 January, 1 March, and 9 April in 2019. The time differences between the three were 37 days and

39 days, respectively. Figure 3 below is the three SPOT6 optical satellite images selected for this study. The SPOT6 satellites are easy to attain in Taiwan, and they are already atmospherically and geometrically corrected. They are completed by the Central University Space Remote Sensing Center. Hence, this study decided to use them. However, in this study, we only used the three indicators of G, R, and IR to generate indicators of ancillary information. For detailed indicator descriptions, please see the content in Section 2.2.

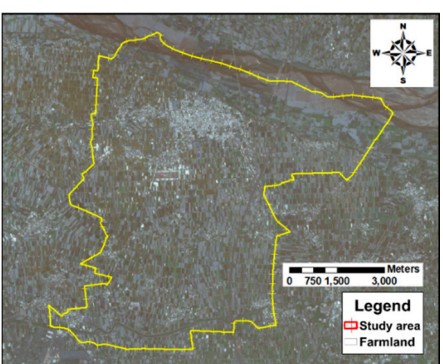 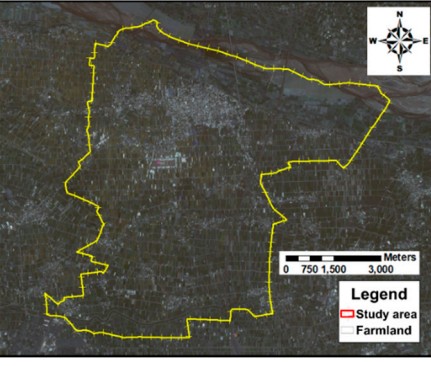 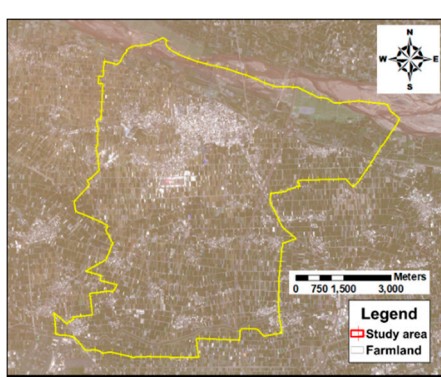

**Time: 23 January 2019**            **Time: 1 March 2019**            **Time: 9 April 2019**

**Figure 3.** The selected SPOT6 satellite image.

Sentinel-1A Images

Sentinel-1A has a shooting period of 12 days, and Sentinel-1 is equipped on four sensors for different shooting purposes, namely Stripmap Mode (SM) and Interfero-metric Wide Swath Mode (IW), Extra Wide Swath Mode (EW), and Wave Mode (Wave Mode, WM). Its spatial resolution is $5 \times 20$ m ($16 \times 66$ feet). To have a better understanding of the rice growth cycle time of this survey, the short time of Sentinel-1 radar image material in this study includes all radar images between 31 January and 7 May 2019. Since the radar images are 12 days old during the shooting cycle, nine radar images during this period are finally selected. All the data can be downloaded from the European Space Agency (ESA) website for free [23]. The downloaded images must be downloaded through the SNAP software. There are three pre-processing steps: radiometric correction, geometric correction, and image speckle noise removal. The selection of radar images was taken on 1/31, 2/12, 2/24, 3/8, 3/20, 4/1, 4/13, 4/25, 5/7. This study uses images in IW mode because this mode is the main operating mode for shooting on land. The shot type is IW and VV polarization and VH polarization images.

### 2.2. Research Design

In this study, 3 of SPOT6 optical satellites and 9 of Sentinel-1A synthetic aperture radar images were used (total 12 images) to identity features such as textures and indicators. The feature value information was extracted using patches as the smallest unit. In the meantime, the time-series features were constructed. Dynamic Time Warping was used to produce "multi-scale time series feature similarity indicators". DTW is dynamic programming [10,11]. The algorithm of this approach compares two sequences with different lengths. It effectively solves the deviation of time distortion in identification and calculates the Euclidean distance between the two sequences to determine the similarity of content information. The gap between the vector distances is rationally found. This indicator can convert an image in a time series of feature information on different scales and different sources into ancillary information.

Figure 4 outlines steps for research, and it has four parts: 1, image feature calculation and database construction; 2, classification algorithm; 3, comparison of classification results. The contents are as follows.

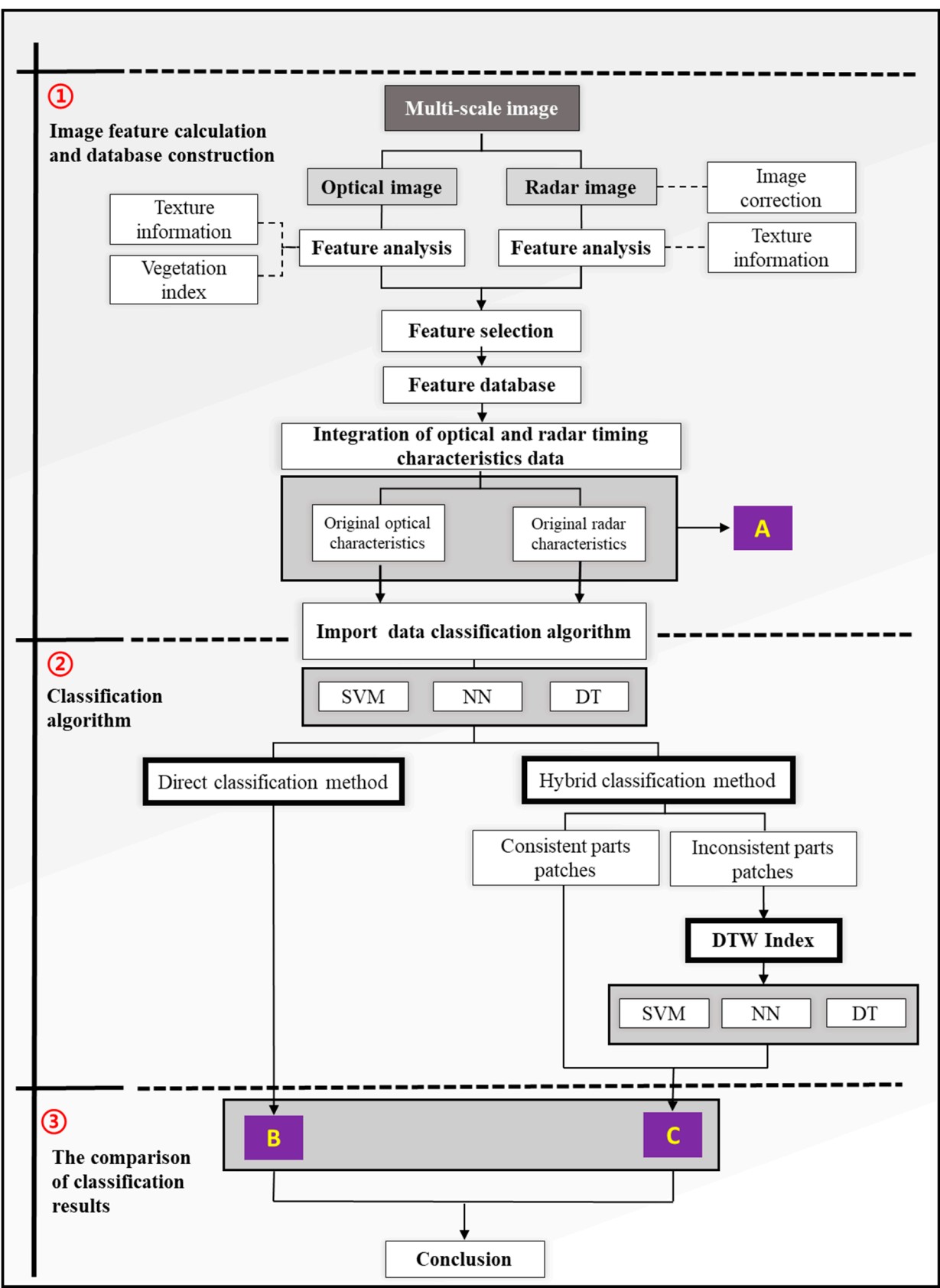

**Figure 4.** Steps for this study.

1.  Image feature calculation and dataset construction: In the optical image part, in addition to the four basic bands of red light (Red), green light (Green), blue light (Blue), and near-infrared light (NIR), this research included Ratio Vegetation Index (RVI) and Normalized Difference Vegetation Index (NDVI) and four Co-occurrence matrix (Gray Level Co-Occurrence Matrix, GLCM) texture indexes, including Homogeneity, Contrast, Dissimilarity, and Entropy, making a total of 19 types of feature information. It is worth mentioning that GLCM and associated texture features are image analysis techniques. An image is composed of pixels, each with an intensity (a specific gray level) suitable to apply GLCM, as different combinations of gray levels often co-occur in an image or image section. Texture feature calculations use the contents of GLCM to measure the variation in intensity (image texture) at the pixel of interest; on the other hand, the radar part uses C-band synthetic aperture radar (SAR) of two polarized images, called VV and VH. In addition to VV and VH, the 4 aforementioned kinds of texture information were also adopted. There was a total of 10 types of feature information. The features are shown in Table 2. In addition, the variation of satellite images for different time features could become a key factor when observing paddy rice and non-paddy rice patterns. We assign this part of the data as Label A.

2.  Classification algorithm: This study used Support Vector Machine (SVM), Neural Network (NN), and C5.0 Decision Tree (DT) machine learning algorithm models. We conducted the training and verification of the models. The model training and verification rate ranged from 70% (37,227 patches) to 30% (15,985 patches). This further illustrates that the design of this research is different from the traditional method. We applied the three classification methods with two direct classification methods and a hybrid classification method to perform better classification outcomes. Direct classification methods use three machine learning models to directly classify the image data information (optical, index, and texture) features (Label A data set). Generally speaking, paddy rice is a long-period crop that requires an indicator and involves the combination of optical, index, and texture features for recording the variation over a long period. Hence, this study adopted two stages to solve the problem. The first stage was to import the image data to three classifiers (SVM, DT, and NN). The second stage was to extract the confusion samples (or patches) from the first stage to employ a new factor (DTW), which considers the time variation factor to improve classification performance. Specifically, this new ancillary information (DTW) was used for the dynamic calculation of the two-time series, and Euclidean distance matrices were computed one by one. For example, optical image characteristic band B to band G is one group, band B to band R is another group, etc. All combinations had to be calculated. The total number of feature information groups reached 26 levels. It was necessary to add 351 combinations of optical feature similarity. The similarity index between the two-time series features was produced, and the dataset was consolidated by the combination of the time series in patch features. The 351 combinations contained Optical Image Feature Similarity (171 attributes), Radar Image Feature Similarity (28 attributes), and the two combinations of feature similarity indicator (optical image + radar image) groups (152 at-tributes). Therefore, in this study, all data combinations were generated to form a DTW index (Figure 5). We hope this process can resolve the confusion around classification patches (samples). In the meantime, the inconsistent patches from the classification model were further refined.

3.  The comparison of classification: The training model accuracy had two parts. The first part was the result of the direct classification method, which was assigned as Label B. The second part was the result of the hybrid classification, which was assigned Label C. The comparison items were computed from an analysis of commission errors and omission errors. The overall accuracy and kappa values were also employed. We also compared the performance of Label B and Label C.

**Table 2.** Ancillary Information.

| Vegetation Index | Formula |
|---|---|
| RVI <br> (Ratio Vegetation Index) | $\frac{R}{NIR}$ |
| NDVI <br> (Normalized Difference Vegetation Index) | $\frac{NIR-R}{NIR+R}$ |
| PVI <br> (Perpendicular Vegetation Index) | $\frac{NIR-NIR_{soil}}{\sqrt{1+B^2}}$ |
| SAVI <br> (Soil-adjusted Vegetation Index) | $(1+L) \times \frac{NIR-R}{NIR+R+L}$ |
| TSAVI <br> (Transformed Soil-adjusted Vegetation Index) | $\frac{B(NIR-NIR_{Soil})}{R+B(NIR-A)+X(1+B^2)}$ |
| CMFI <br> (Cropping Management Factor Index) | $\frac{R}{NIR+R}$ |
| GI <br> (Greenness Index) | $\frac{NIR}{G}$ |
| IPVI <br> (Infrared Percentage Vegetation Index) | $\frac{NIR}{NIR+R}$ |
| MSAVI <br> (Modified Soil-adjusted Vegetation Index) | $\frac{\left(2NIR+1-\sqrt{(2NIR+1)^2-8(NIR-R)}\right)}{2}$ |
| OSAVI <br> (Optimization Soil adjusted Vegetation Index) | $\frac{NIR-R}{NIR+R+Y}$ |
| GESAVI <br> (Generalize Soil- adjusted Vegetation Index) | $\frac{NIR-NIR_{Soil}}{R+Z}$ |
| HOM <br> (Homogeneity) | $Homogeneity = \sum\limits_{i=0}^{N} \sum\limits_{j=0}^{N} \frac{1}{1+(i-j)^2} C_{ij}(d,\theta)$ |
| CON <br> (Contrast) | $Contrast = \sum_{i,j} |i-j|^2 p(i,j)$ |
| DIS <br> (Dissimilarity) | $Dissimilarity = \sum\limits_{i=0}^{n} \sum\limits_{j=0}^{n} C_{ij}|i\,j|$ |
| ENT <br> (Entropy) | $Entropy = \sum\limits_{i=0}^{n} \sum\limits_{j=0}^{n} C_{ij} \log C_{ij}$ |

Experienced Coefficient for $L = 0.5$; $X = 0.08$; $Y = 0.16$; $Z = 0.35$ considering multiple scattered conditions: $R_{Soil} = A + B \times R$; ($A = 0.011$, $B = 1.16$) [24]. $NIR_{Soil} = BR - A$; The $A$ and $B$ is the soil line parameters. BR = Blue light $\times$ $B$.

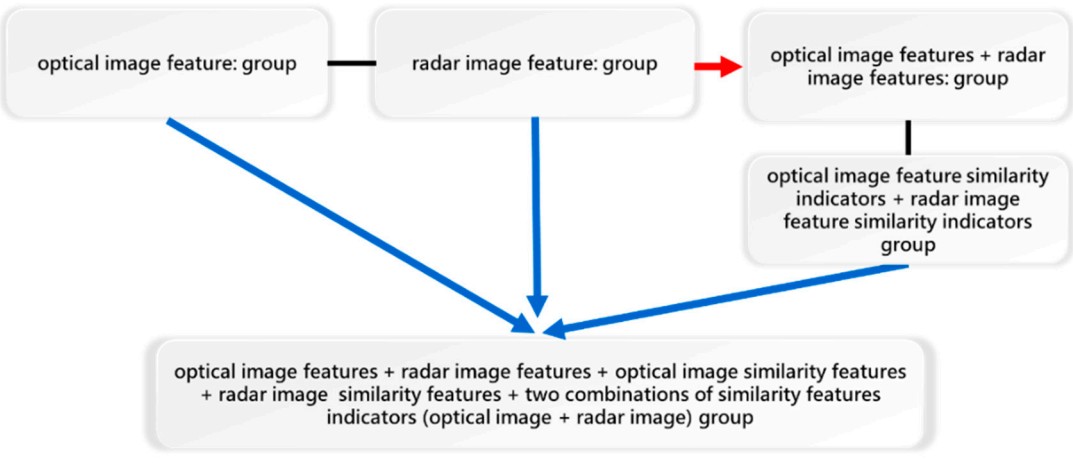

**Figure 5.** The DTW index of analyzed data sets for steps.

## 3. Research Methods

### 3.1. Support Vector Machine

The Support Vector Machine (SVM) is a popular machine learning tool that offers solutions for both classification and regression problems. Support vector machines (SVMs) are well-accepted supervised learning methods used for classification. The SVM classifier supports binary classification and multiclass classification, whereas the structured SVM trains a classifier for generally structured output labels. Moreover, there exist many hyperplanes that may be able to classify the data. One rational choice as the best hyperplane is to produce the largest separation, or margin, between the two classes. The optimal choice of the hyperplane is the distance from the selected sample to the nearest sample point on each side, which is maximized. The study considers the concept of improving statistical learning theory, generally applied as an effective classifier to solve many practical problems. The feature of these classifiers is to minimize the empirical classification error and maximize the geometric margin [25]. The support vector machine is requested to select an appropriate kernel function. The function of the kernel is to take data as input and convert it into the required form. This is because different types of data cannot be linearized in the original space. When separated, the data after nonlinear projection can be more separated in a higher-dimensional space, usually linear, polynomial, Radial Basis Function (RBF), and Sigmoid Function. We explain the classification methods of SVM in detail. The value of bias is set to 0. The core function is adopted as RBF. The c = 10 and gamma = 0.1 are used as the initial conditions for setting parameters. The stopping criterion of 0.001 was used as a standard to terminate the program and output the results.

### 3.2. Neural Network

The neural networks are information processing networks inspired by the way biological neural systems process data. Neural Networks (NN) were first proposed in the early 1940s as an attempt to simulate human brain cognitive learning processes [26]. They are programmed with a primary function, which is to develop models of problems based on trial and error or learning procedures. In the last decades, Back Propagation Neural Network has been widely applied in many fields. The relations among massive data and a certain phenomenon are obtained through a learning system (instead of calculation). In the past, scientists and researchers experienced that the inputs of attributes (included ancillary information) for remote sensing images are usually used to apply to image classification. If a paddy area spatial dataset was well developed to perform the input variables and output categories rationally, it may be appropriate to apply Back Propagation Neural Network as a learning machine [27]. Basically, the neural network consists of many nodes to connect input neurons and output neurons to three sorts of layers: input layer, hidden layer, and output layer. The study adopted Multi-Layer Perception. Our MLP consists of at least three layers of nodes: an input layer, a hidden layer with 13 neurons, and an output layer of classification. The optical input has 19 neurons, and the radar input has 8 neurons. The active function used the sigmoid function. The output used 800 epochs or 0.02% difference as a criterion to obtain the classification outcomes.

### 3.3. Decision Tree

A decision tree is a tree structure containing internal and external nodes connected by branches. A decision tree is a data-driven predictive model where it is mapped from the observation of samples about an item to conclusions about its target value. It is usually used as a tool for scientists and engineers to generate "rules" [28]. The internal node is a decision-making point to investigate a decision function to determine which child node to visit next. On the other hand, an external node is also known as a leaf or terminal node, which has no child nodes and, with respect to a label, characterizes the given data that lead to being visited. In general, a decision tree is employed as follows. It presents a datum (a vector composed of several attributes) to the root node of the decision tree. It may depend on the result of a decision function used by an internal node, and the tree will

branch to one of the children of the node. This process will repeat until a terminal node is approached and a label or value is then assigned to the given data. The height of DT is limited to 17 layers, which uses the Exhaustive Algorithm to display all the possibilities of the condition the samples should fit in.

### 3.4. DTW Methods

DTW is one of the algorithms for computing the similarity on two temporal sequences. Hence, DTW can be successfully applied to temporal sequences of video, audio, and graphics data. That is, any data that can be turned into a linear sequence can be analyzed by DTW as well. A well-known application has been automatic speech recognition for different speaking speeds. It can also be used in partial shape matching applications. Reviewing the DTW past research, Petitjean et al. (2012) used the DTW algorithm for SPOT time-series satellite images to classify the land-use coverage. This research incorporated the K-means and DTW into image measurement to obtain the classification. The results show that the similarity of multi-period images is matched by using DTW, which performs better in classification outcomes of multi-period images than that of a single image use [10].

Similarity measurements between the two sequences are named as "warping path". In this path the two signals may be aligned at the same time. The signal with an original set of points X (original), Y (original) is converted to X (warped), Y (warped). Related technique sequences of varying speed may be averaged using this technique. While two different time series data are matched with each other, it can be seen directly through a line chart or other visualized graphs whether there is a strong similarity between the two. It is possible to objectively quantify the degree of similarity between the two images.

To calculate the DTW similarity of two-time series, one can establish an m × n Euclidean distance matrix. Then, a cost matrix or cumulative matrix $M_c$ based on the distance matrix is generated. The cumulative matrix is $M_c$ $(i, j)$ defined as follows:

$$M_C(i,j) = \min \begin{cases} M_C(i-1, j-1) \\ M_C(i-1, j) \\ M_C(i, j-1) \end{cases} + M(i,j) \tag{1}$$

In Equation (1), $M_C(i,j)$ represents a matrix from a point $(i, j)$ of the route. The accumulated value of the minimal value in (1) has three terms. The optimal value can be found by considering $M_C(i,j)$ with two periods of distance in DTW by computing from $(1, 1)$ to $(i, j)$.

Since the DTW can analyze two sets of timing information with different scales and timing lengths, it produces the most intuitive numerical value to show the degree of similarity of the timing fluctuations between the two sets of information [10–22]. This study presumes that different sources of information have their contributions for classification. We utilized the "multi-scale time series feature similarity" indicators from the concept of data fusion, especially considering the ancillary information of radar data, to compare the optical image data to produce the similarities. According to Equation (1), this study uses Python to write a "multi-scale time series feature similarity indicators" program that can process multiple time series feature information in batches. The program merges all the characteristics into it as well. The features are computed as a feature similarity index, and then each of the rest of the features step-by-step will be imported. All the aforementioned data are applied to the classification outcomes from three approaches (SVM, NN, and DT) by inconsistent classified results. The best way to resolve the inconsistent classified results is to use new ancillary information (such as DTW).

### 3.5. Accuracy Verification

To test the accuracy of the final automated classification model, this study uses the Confusion Matrix and kappa value in image interpretation and classification accuracy of the final results of this research. Four different regions are randomly selected in this study area as verification regions to check the final results of the developed new classification model.

### 3.6. Model Software

In this study, we used IBM SPSS Modeler 18.1 to carry out the analysis. The software is user-friendly with graphical interference to display how the outcome is obtained.

## 4. Results and Discussion

### 4.1. Examples of Optical and Radar Timing Characteristics Data

The results of this research are shown in Label A in Figure 4. Conventionally, vegetation indices and texture information can successfully classify paddy rice through image classification. However, this study made further progress. Taking a closer look at the radar image, Figure 6 shows the texture characteristic curve of Sentinel-1A (VH) and (VV) polarization images, respectively. Overall, entropy and homogeneity display dramatic differences in the time series analysis for paddy rice and non-paddy rice. In addition, rice and non-rice also show differences among these indicators. By carefully examining the texture analysis of Figure 6a,c in the changes of rice growth, it can be found that rice transplanting (transplanting rice seedlings) happens after 31 January. While the rice grows, the rice leaves gradually cover the surface soil. Then the leaves continuously have edges broken, and the bright spots and flat areas increase at the same time. The changes in the entire texture information are inconsistent, and the texture value decreases. After paddy heading from 24 February to 1 April as the ears of rice grow, the degree of texture disorder (see the entropy indicator) gradually increases, while the homogeneity decreases slightly and the homogeneity area produces a smaller value. From 1 April to 25 April as the rice leaves grow to cover the soil reflection, the texture tends to be consistent. The entropy decreases while the homogeneity increases. On 27 May when the rice ears are mature and exposed, the paddy harvest period begins. The rice ears of these highly reflective objects reduce the uniformity, so the texture value increases sharply [29]. According to the analysis of the aforementioned waveband information, we notice certain temporal characteristics of the trend information t. In the past, we tended to ignore these changes in the time axis. In Figure 6, the x axis is the observation time, and y axis is the normalization value of each type of texture information. Previously, there was a lack of an ideal tool to effectively integrate the information. If we employ the subsequent classification algorithms to increase the effective image information, we can certainly provide some help for the classification of rice fields.

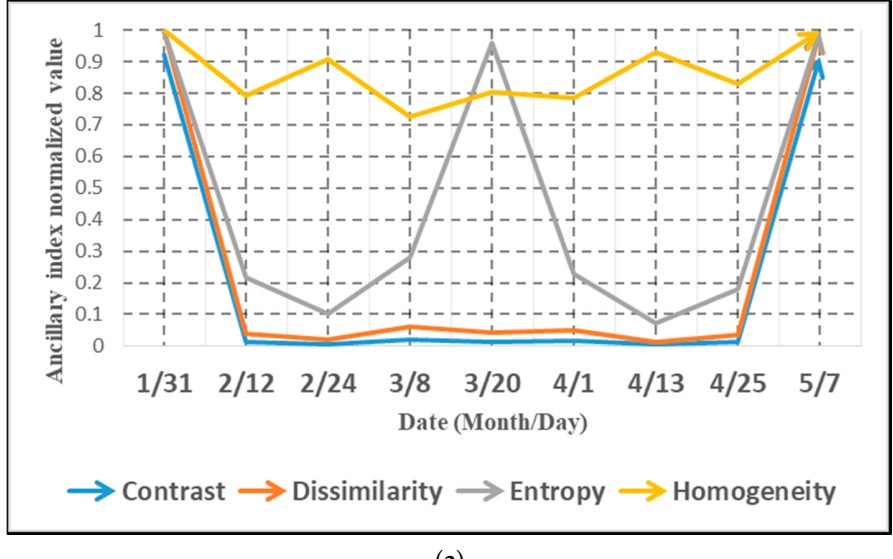

(**a**)

**Figure 6.** *Cont.*

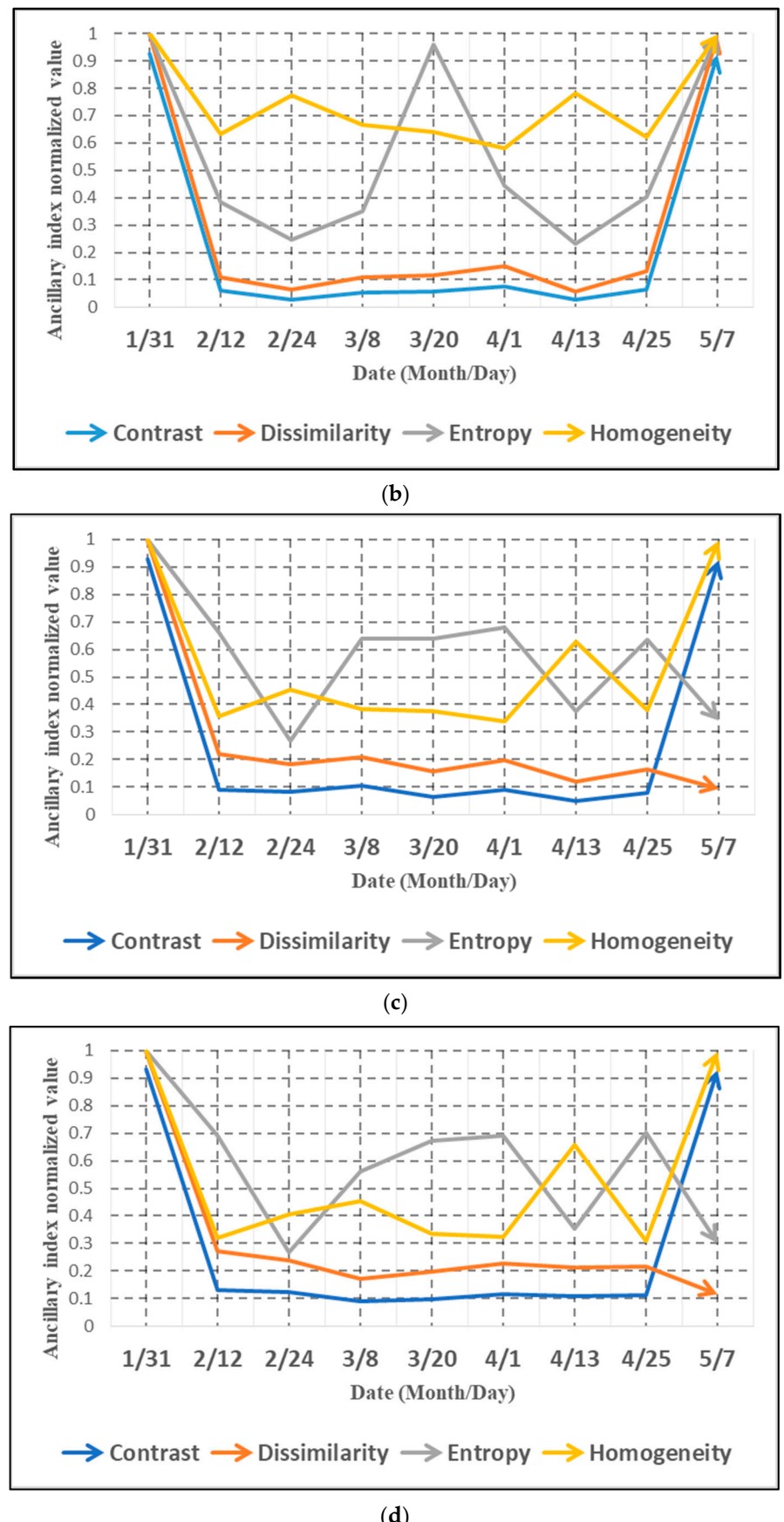

**Figure 6.** Sentinel-1A VH and VV variation, (**a**) paddy rice (VH), (**b**) non-paddy rice (VH), (**c**) paddy rice (VV), and (**d**) non-paddy rice (VV).

### 4.2. Comparison on Direct Classification Method and Hybrid Classification Method

Since the range of study area is too large, we chose a small area to display the classification performance result. However, the confusion matrix is generated by the entire study area. The range is presented as the red frame in Figure 7.

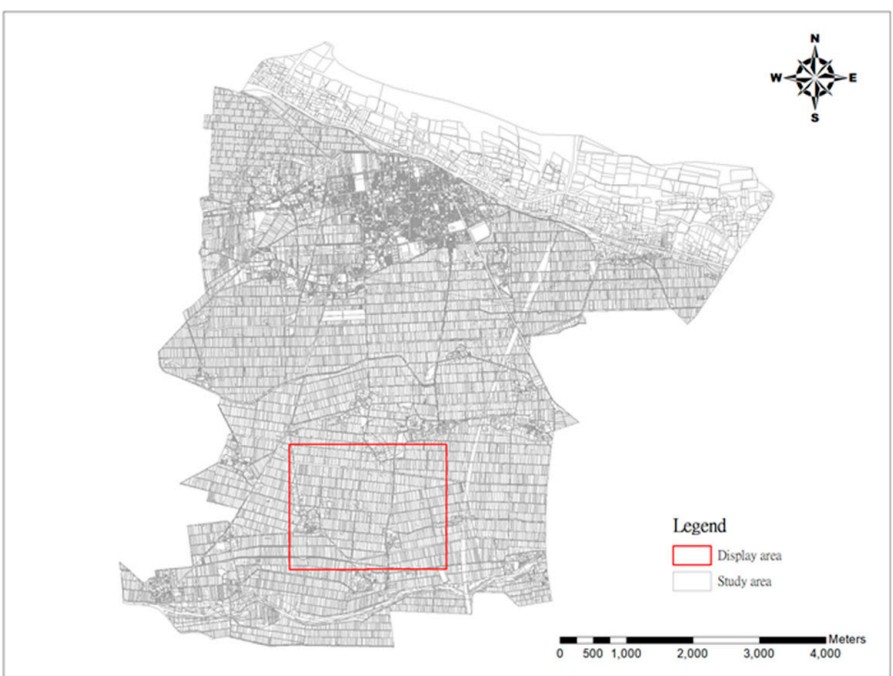

**Figure 7.** Display area with detailed information.

### 4.2.1. Direct Classification Method

The results of this research are shown in Label B in Figure 4. Table 3 shows the analysis results of the direct classification methods of this study. From top to bottom, the methods are SVM, NN, and DT. Among the three classification methods, DT achieved the best accuracy and kappa value of 93.26% and 0.76, respectively. The worst result was NN; overall accuracy and kappa value were, respectively, 89.51% and 0.66. It seems that the overall median value is the SVM method. Figure 8 shows the results displayed by the three algorithms. The blue frame in Figure 8 indicates that the calculation result is inconsistent compared to the ground truth data. In terms of direct classification, no matter which algorithm we used, the commission errors of rice still accounted for a certain number of cases.

However, we decided to examine how to integrate the optical information and radar information by considering multiple data resources with multiple algorithms. See Table 3 for further information. For this algorithm, the commission error of rice is quite serious. Although a large number of texture images in the analysis process are used, it seems that there are still many commission errors in the classification problem. Even if we use radar information at the same time, this does not seem to enhance the performance. On the other hand, this shows that the current analysis results may tend to be over-trained in the non-rice part. There are many reasons for the result of rice misclassification. This reason is a common phenomenon in the problem of rice classification because the rice samples are grown on the ground in different time scenarios. To solve the problem of commission errors, we employ a hybrid classification method in the next step.

**Table 3.** Direct classification.

| SVM | | Ground Truth | | | Producer's Accuracy |
|---|---|---|---|---|---|
| | | **Paddy Rice** | **Non-Paddy Rice** | **Sum of Columns** | |
| Direct Classification | Paddy Rice | 7173 | 3868 | 11,041 | 0.65 |
| | Non-Paddy Rice | 526 | 41,645 | 42,171 | 0.99 |
| | Sum of Rows | 7699 | 45,513 | 53,212 | |
| | User's Accuracy | 0.93 | 0.92 | | |
| | Accuracy | 91.74% | | | |
| | kappa | 0.72 | | | |
| NN | | Ground Truth | | | Producer's Accuracy |
| | | **Paddy Rice** | **Non-Paddy Rice** | **Sum of Columns** | |
| Direct Classification | Paddy Rice | 7075 | 4958 | 12,033 | 0.59 |
| | Non-Paddy Rice | 624 | 40,555 | 41,179 | 0.99 |
| | Sum of Rows | 7699 | 45,513 | 53,212 | |
| | User's Accuracy | 0.92 | 0.89 | | |
| | Accuracy | 89.51% | | | |
| | kappa | 0.66 | | | |
| DT | | Ground Truth | | | Producer's Accuracy |
| | | **Paddy Rice** | **Non-Paddy Rice** | **Sum of Columns** | |
| Direct Classification | Paddy Rice | 7250 | 3139 | 10,389 | 0.70 |
| | Non-Paddy Rice | 449 | 42,374 | 42,823 | 0.99 |
| | Sum of Rows | 7699 | 45,513 | 53,212 | |
| | User's Accuracy | 0.94 | 0.93 | | |
| | Accuracy | 93.26% | | | |
| | kappa | 0.76 | | | |

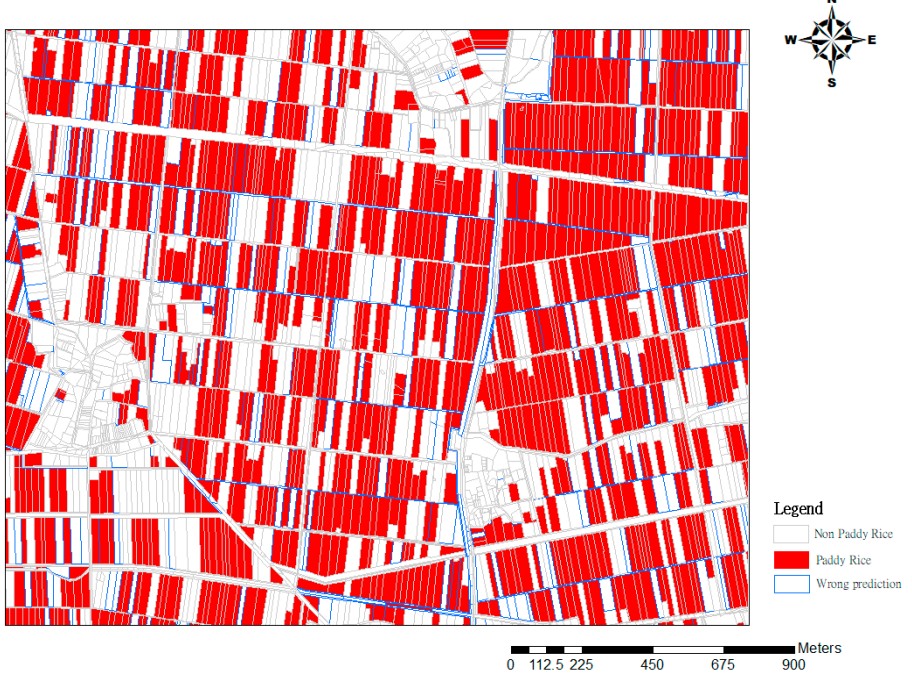

(**a**)

**Figure 8.** *Cont.*

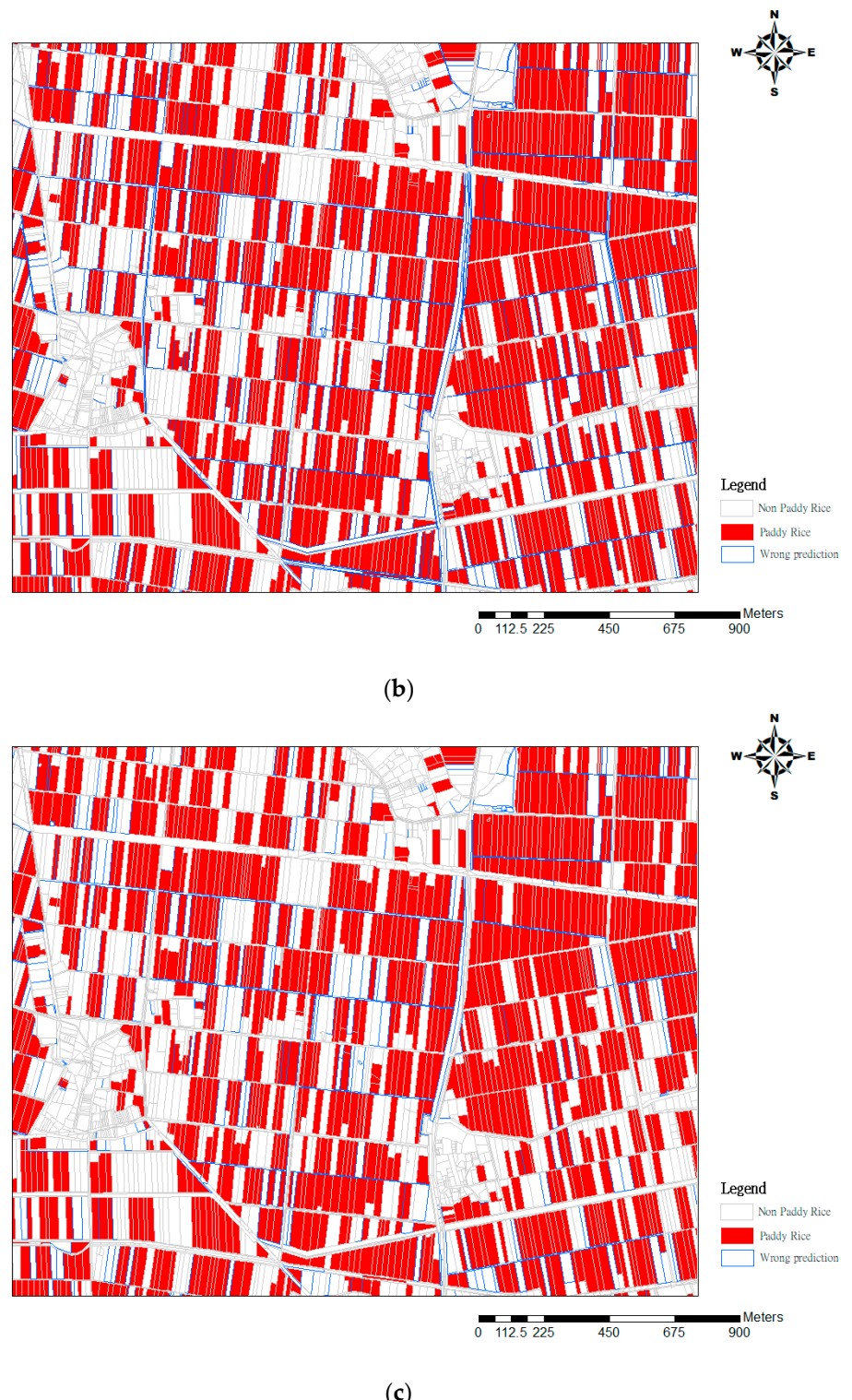

(**b**)

(**c**)

**Figure 8.** Comparison of direct classification and ground truth data, (**a**) SVM, (**b**) NN, and (**c**) DT.

4.2.2. Hybrid Classification

The results are shown in Label C in Figure 4. The hybrid classification method is divided into two parts. In the first, the patches show consistency after the first stage of classification, regardless if they show rice or non-rice (49,084 patches). The number of patches with inconsistent parts is 4128, so we execute re-classification. In the meantime, the DTW index information of "multi-scale time series feature similarity indicators" is employed. The DTW is calculated based on the values for the entire image, and the

extraction of the inconsistent patches is carried out individually. Then, they are newly plugged into the dataset and re-classified. Therefore, we explain the results of the three parts as follows: 1, the accuracy of consistency classification; 2, inconsistency classification patches for discussion; 3, DTW indicator results; 4, hybrid classification method integration accuracy results.

Step 1. The first stage of accuracy of consistency classification.

This study uses a two-stage classification. Showing the results of the first stage of the multi-calculation classification method, Table 4 presents the patches that were consistent in the classification among the three algorithms. The meaning of this analysis is that current input feature variables for classification reach the best limitation of classification. In other words, considering a classification model under the condition of maximizing the accuracy of image data, the maximum classification approach of the machine learning model may be found.

**Table 4.** Consistency of classification outcome.

| | | Ground Truth | | | Producer's Accuracy |
|---|---|---|---|---|---|
| | | **Paddy Rice** | **Non-Paddy Rice** | **Sum of Columns** | |
| Consistency of Classification | Paddy Rice | 6904 | 2208 | 9112 | 0.76 |
| | Non-Paddy Rice | 292 | 39,680 | 39,972 | 0.99 |
| | Sum of Rows | 7196 | 41,888 | 49,084 | |
| | User's Accuracy | 0.96 | 0.95 | | |
| | Accuracy | 94.91% | | | |
| | kappa | 0.82 | | | |

Step 2. Discuss accuracy of consistency classification.

Table 5 presents the patches that were inconsistent in the classification among the three algorithms. From top to bottom, they are SVM, NN, and DT. Among these three classification methods, DT had the best accuracy and kappa value of 73.64% and 0.26, respectively. The worst result was NN. Its overall accuracy and kappa value were, respectively, 25.34% and −0.14. Table 5 shows that the patches had inconsistent classifications under different classification approaches. Both rice and non-rice samples had extreme commission errors and omission errors. Furthermore, there are many reasons for the resulting commission errors of rice. The complications of image quality and planting methods (time difference, mixed planting) are the major reasons. It is very difficult to resolve them by using an existing classifier unless time-history data are employed. Hence, in this study, we decided to incorporate DTW and time feature variables to provide three algorithms for the second stage classification. According to the results when employing DTW in the classification process, the proposed approach enhanced classification to the maximum classification level.

**Table 5.** Inconsistency of classification outcome.

| **SVM** | | Ground Truth | | | Producer's Accuracy |
|---|---|---|---|---|---|
| | | **Paddy Rice** | **Non-Paddy Rice** | **Sum of Columns** | |
| Inconsistency of Classification | Paddy Rice | 269 | 1660 | 1929 | 0.14 |
| | Non-Paddy Rice | 234 | 1965 | 2199 | 0.89 |
| | Sum of Rows | 503 | 3625 | 4128 | |
| | User's Accuracy | 0.53 | 0.54 | | |
| | Accuracy | 54.12% | | | |
| | kappa | 0.03 | | | |

**Table 5.** *Cont.*

| NN | | Ground Truth | | | Producer's Accuracy |
| --- | --- | --- | --- | --- | --- |
| | | **Paddy Rice** | **Non-Paddy Rice** | **Sum of Columns** | |
| Inconsistency of Classification | Paddy Rice | 171 | 2750 | 2921 | 0.06 |
| | Non-Paddy Rice | 332 | 875 | 1207 | 0.72 |
| | Sum of Rows | 503 | 3625 | 4128 | |
| | User's Accuracy | 0.34 | 0.24 | | |
| | Accuracy | 25.34% | | | |
| | kappa | −0.14 | | | |
| DT | | Ground Truth | | | Producer's Accuracy |
| | | **Paddy Rice** | **Non-Paddy Rice** | **Sum of Columns** | |
| Inconsistency of Classification | Paddy Rice | 346 | 931 | 1277 | 0.27 |
| | Non-Paddy Rice | 157 | 2694 | 2851 | 0.94 |
| | Sum of Rows | 503 | 3625 | 4128 | |
| | User's Accuracy | 0.69 | 0.74 | | |
| | Accuracy | 73.64% | | | |
| | kappa | 0.26 | | | |

Step 3. Examples of multi-scale time series feature similarity indicators and description of integration results.

Table 6 presents a dataset that converts the time series dynamic relationship of the information. That is, each patch is generated by an index for the calculation of "multiscale time series feature similarity indicators" in this research. Because the amount of data is too large, we extracted a part of the data to present the research results.

**Table 6.** The relations of patches and multi-scale of features.

| Patch number | The number of multi-scale of feature on similarity index | | | | | | | | | | | | | |
| --- | --- | --- | --- | --- | --- | --- | --- | --- | --- | --- | --- | --- | --- | --- |
| | 1 | 2 | 3 | 4 | 5 | 6 | 7 | 8 | 9 | 10 | ...... | 349 | 350 | 351 |
| 1 | 0.1027 | 0.7318 | 0.2544 | 0.0869 | 0.0868 | 0.0870 | 0.0872 | 0.0874 | 0.0874 | 0.0874 | | 0.0124 | 0.0044 | 0.0088 |
| 2 | 0.1319 | 0.5630 | 0.1165 | 0.1041 | 0.1039 | 0.1043 | 0.1046 | 0.1049 | 0.1049 | 0.1049 | | 0.0104 | 0.0016 | 0.0078 |
| 3 | 0.0448 | 0.4065 | 0.0985 | 0.2306 | 0.2305 | 0.2310 | 0.2312 | 0.2316 | 0.2315 | 0.2315 | | 0.0028 | 0.0017 | 0.0029 |
| 4 | 0.0901 | 0.2802 | 0.0601 | 0.2640 | 0.2639 | 0.2644 | 0.2646 | 0.2651 | 0.2649 | 0.2649 | | 0.0084 | 0.0015 | 0.0064 |
| 5 | 0.0892 | 0.4852 | 0.1634 | 0.1190 | 0.1188 | 0.1193 | 0.1194 | 0.1196 | 0.1196 | 0.1196 | | 0.0020 | 0.0035 | 0.0030 |
| 6 | 0.2003 | 0.6422 | 0.2344 | 0.0276 | 0.0276 | 0.0278 | 0.0280 | 0.0284 | 0.0283 | 0.0283 | | 0.0011 | 0.2824 | 0.2861 |
| 7 | 0.0974 | 0.4458 | 0.1200 | 0.1045 | 0.1044 | 0.1047 | 0.1049 | 0.1053 | 0.1052 | 0.1052 | | 0.0020 | 0.3954 | 0.4006 |
| 8 | 0.0784 | 0.4373 | 0.0988 | 0.1061 | 0.1060 | 0.1064 | 0.1066 | 0.1069 | 0.1068 | 0.1068 | | 0.0022 | 0.1980 | 0.2008 |
| 9 | 0.0598 | 0.4317 | 0.1187 | 0.1217 | 0.1216 | 0.1219 | 0.1221 | 0.1224 | 0.1223 | 0.1223 | | 0.0037 | 0.3297 | 0.3345 |
| 10 | 0.0870 | 0.4811 | 0.1436 | 0.1090 | 0.1089 | 0.1093 | 0.1095 | 0.1098 | 0.1098 | 0.1098 | | 0.0031 | 0.4233 | 0.4297 |
| ⋮ | | | | | | ...... | | | | | | | | |
| 53210 | 0.0545 | 0.2967 | 0.0845 | 0.1275 | 0.1274 | 0.1278 | 0.1279 | 0.1282 | 0.1282 | 0.1282 | | 0.0148 | 0.0157 | 0.0062 |
| 53211 | 0.0920 | 0.4646 | 0.1588 | 0.0749 | 0.0748 | 0.0752 | 0.0753 | 0.0756 | 0.0755 | 0.0755 | | 0.0113 | 0.0116 | 0.0042 |
| 53212 | 0.1268 | 0.4366 | 0.1357 | 0.0902 | 0.0900 | 0.0905 | 0.0906 | 0.0909 | 0.0909 | 0.0909 | | 0.0046 | 0.0052 | 0.0037 |

In Table 6, the y axis is the number of patches in the demonstration area, including number of 53,212. The x axis is the number of combinations of multi-scale time series features, i.e., 351 features, and each grid value in Table 6 is the time series between the two-time series features of different patches. The higher the similarity value is, the more similarity between them. For instance, taking patch ID = 1 as an example, one to four groups of feature groups of similar value indexes are sorted by size as follows: feature dataset 2 (0.7318) > dataset 3 (0.2544) > dataset 1 (0.1027) > dataset 4 (0.0869). The information of the feature number is derived from dataset one to four as (1) Dataset1: SPOT6 red light band vs. SPOT6 green light; (2) Dataset2: SPOT6 red light band vs. SPOT6 blue light; (3) Dataset3: SPOT6 red light band vs. SPOT6 near-infrared light; (4) Dataset4: SPOT6 near-infrared light vs. crop management factor index (CMFI).

The above example shows the calculation of this indicator. The similarity between time series features of different scales can be converted into actual values, and the dataset's multiple features can be integrated into a worksheet, which greatly increases the analysis among different time domains and various data sources. It is worth mentioning that our analysis at this stage analyzes the entire image. The above analyses are easy to govern numerically because we can trace the IDs for their corresponding locations.

Step 4. The overall accuracy of the hybrid classification.

Table 7 shows the analysis results of the hybrid classification method of this study. The results are presented for SVM, NN, and DT. From the results, when we compare the three classification methods, DT had the best classification result. The accuracy and kappa values were 94.71% and 0.81, respectively. NN showed the worst result again, with overall accuracy and kappa values of 92.63% and 0.74, respectively. The NN approach still greatly improved the classification accuracy when applying DTW. The other two classifiers (SVM and DT) also had increased performance in classification for DTW. The final classification results of DT and SVM are largely the same. This also shows that the new ancillary information of DTW can sustainably improve the classification results. Figure 9 shows the results of the three algorithms. By zooming in on the selected area, it can be found that the original direct classification method achieved more significant improvement in commission errors than in omission errors. Figure 9 shows that there are many yellow frames indicating correction of NN, which also means that the prediction accuracy of NN in the hybrid classification method is improved when compared to the direct classification method. In other words, the result shows that the DTW indicator can provide better classification performance. To sum up, we decided to employ the DTW index in the classification process. Our results show how the DTW index resolved the confusing parts of the image. As usual, if a pixel is classified as the same pattern by different classifiers, very few errors are produced [9]. However, if a pixel is not classified as the same category for a different classifier, deployment of a new indicator (DTW) can be expected to update the erroneous pattern.

**Table 7.** Outcomes for hybrid classification.

| SVM | | Ground Truth | | | Producer's Accuracy |
|---|---|---|---|---|---|
| | | **Paddy Rice** | **Non-Paddy Rice** | **Sum of Columns** | |
| Hybrid Classification | Paddy Rice | 7360 | 2626 | 9986 | 0.74 |
| | Non-Paddy Rice | 339 | 42,887 | 43,226 | 0.99 |
| | Sum of Rows | 7699 | 45,513 | 53,212 | |
| | User's Accuracy | 0.96 | 0.94 | | |
| | Accuracy | 94.43% | | | |
| | kappa | 0.80 | | | |

**Table 7.** *Cont.*

| NN | | Ground Truth | | | Producer's Accuracy |
|---|---|---|---|---|---|
| | | **Paddy Rice** | **Non-Paddy Rice** | **Sum of Columns** | |
| Hybrid Classification | Paddy Rice | 7184 | 3407 | 10,591 | 0.68 |
| | Non-Paddy Rice | 515 | 42,106 | 42,621 | 0.99 |
| | Sum of Rows | 7699 | 45,513 | 53,212 | |
| | User's Accuracy | 0.93 | 0.93 | | |
| | Accuracy | 92.63% | | | |
| | kappa | 0.74 | | | |

| DT | | Ground Truth | | | Producer's Accuracy |
|---|---|---|---|---|---|
| | | **Paddy Rice** | **Non-Paddy Rice** | **Sum of Columns** | |
| Hybrid Classification | Paddy Rice | 7397 | 2512 | 9909 | 0.75 |
| | Non-Paddy Rice | 302 | 43,001 | 43,303 | 0.99 |
| | Sum of Rows | 7699 | 45,513 | 53,212 | |
| | User's Accuracy | 0.96 | 0.94 | | |
| | Accuracy | 94.71% | | | |
| | kappa | 0.81 | | | |

Usually, there are two ways to express classification accuracy, the first is the overall accuracy (OA), and the second is the kappa value. OA represents the proportion of the number of correctly classified samples to the total number of samples, but such indicators are easily affected by the omission error and commission error rate. Thus, the kappa value must be considered. The kappa is a better reference than OA to observe commission errors and omission errors. For instance, kappa results for SVM (0.72 vs. 0.80), NN (0.66 vs. 0.74), and DT (0.76 vs. 0.81) are adopted in Tables 3 and 7 which proved that the three classifier models are satisfactory in terms of applying DTW.

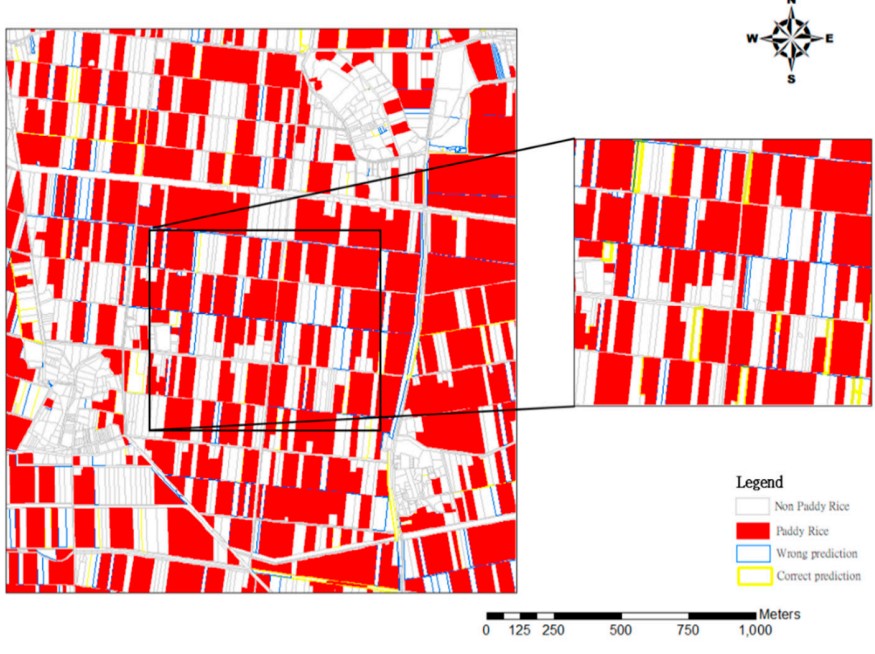

(**a**)

**Figure 9.** *Cont.*

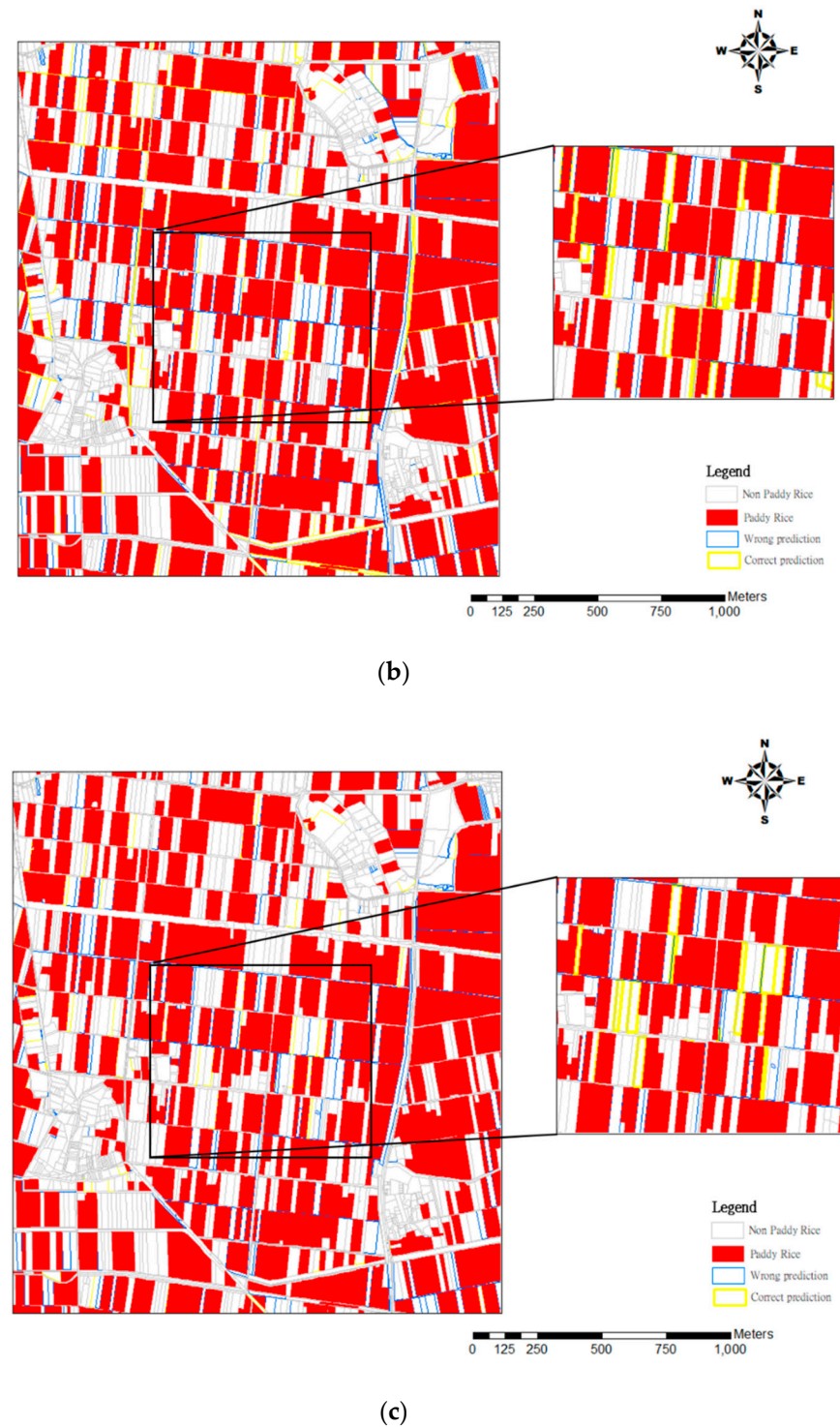

**Figure 9.** Comparison on hybrid classification and ground truth data, (**a**) SVM, (**b**) NN, and (**c**) DT.

Overall, DTW is based on the dynamic programming method for effectively reducing search and comparison time. The multi-scale time series feature similarity indicators developed in this research have the ability to transform multi-dimensional data into two-dimensional information. The reason for applying DTW is because the status of crops is a long-term characteristic. This research shows that time features are helpful for long-term characteristic image classification. In particular, it can be used in small farmland areas and fragile landscapes. Through the integration of DTW data, it can overcome the limitations of the large difference between optical images and radar images. In addition, the different spatial

resolutions of the two types of images are integrated. Moreover, the various limitations of different atmospheric conditions in the shooting process of the two images are resolved. This indicator has extremely high potential in the detection of crop phenology.

## 5. Summary and Conclusions

This study developed a multi-scale time series feature similarity index through the Dynamic Time Wrapping (DTW) theory to integrate multi-source scale time-series image information. The training/test dataset was analyzed through a verification process proving that the original feature information was added to the time series similarity index of the multi-scale time series feature data. The conclusions of this paper are as follows:

1.  This study used SPOT6 optical images and Sentinel-1A radar images as the materials of research, which differs from the mainstream use of image fusion in the interpretation in past studies. The massive time series features in the datasets are integrated into a simple index to present the data dimensions in a single dataset. This approach provides new possibilities for subsequent analysis of information considering different scales of data.

2.  The homogeneity and entropy in radar images provides some new information in time series analysis, which greatly helps the classification of paddy rice. It is found that the behavior of time variations can distinguish paddy rice and non-paddy rice easily.

3.  This study uses the "direct classification method" and "hybrid classification method" for comparison. The characteristic information of optical satellite images and radar images is applied to directly perform classification methods for their behaviors. The results show that the overall accuracy results of the direct classification method are 91.7% (kappa value 0.72, SVM), 89.5% (kappa value 0.66, NN), and 93.26% (kappa value 0.76, DT). In the second stage of classification, the patches were classified optically with DTW feature information using three approaches, and neutral patches were added in the first stage, producing the overall accuracy results of 94.43% (kappa value is 0.80, SVM), 92.63% (kappa value is 0.74, NN), and 94.71% (kappa value is 0.81, DT). This also proves the DTW is robust.

4.  This result renders a feasible way to integrate radar feature information with optical feature information, especially in multi-period data. The optical images in different periods are difficult to obtain due to the influence of weather conditions. Radar images can be obtained regularly since cloud and fog interference can be avoided. A possible solution has been designed to overcome their disadvantages, which could lead to better classification performance. Considering those various restrictions, it is especially suitable for small farmland areas and fragile landscapes.

**Author Contributions:** T.C.L. was responsible for the plan and design of this study. He analyzed the data and discussion. S.W. helped with writing the manuscript and discussion of results. Y.C.W. wrote the computer program. H.-P.W. and C.-W.H. used the program to plot the thematic map and generate tables. All authors have read and agreed to the published version of the manuscript.

**Funding:** This research was funded by the Ministry of Science and Technology (MOST) 103-2119-M-035 -002 –.

**Acknowledgments:** The authors would like to thank the MOST for providing image data and related information. The authors are also very grateful to Z. H. Zhu, Department of Geography, National Taiwan University, for his advice and suggestions for this project.

**Conflicts of Interest:** The authors declare no conflict of interest.

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
