# Peer review of "Multi-Temporal Data Fusion in MS and SAR Images Using the Dynamic Time Warping Method for Paddy Rice Classification"

_agriculture, doi:10.3390/agriculture12010077_

Round 1

Reviewer 1 Report

Dear authors,

After reading your article, I noticed the improvement of the formatting, but no improvement of the content from the previous version. 

I read again the article. English mistakes have been corrected. Point of view of the scientific content, no significant improvement is noticed.  

Author Response

Thank you for your careful reading. We fixed all the errors of values and English in the new version.  The novelty of the paper is in the blue text.

Reviewer 2 Report

I am satisfied with this revised manuscript since the Authors have appreciated and altered the manuscript according to the previous comments.

In my opinion, I would change last Section only to Conclusions since Summary of the manuscript has to be explained/described in the previous Section. And according to the previous comment, this Section should be shortened in a way that it  summarizes overall arguments or findings from the research

Author Response

please see the attached file

We shorten the Conclusion parts.

Round 2

Reviewer 1 Report

Dear authors,

Thank you for the answers and good luck in the future!

This manuscript is a resubmission of an earlier submission. The following is a list of the peer review reports and author responses from that submission.

Round 1

Reviewer 1 Report

After reading the article I have the following remarks.

The study is interesting providing a comparison between different approaches used for paddy-rice classification and proposes a new one, that gave better results than the known ones. Still, given the English inappropriate use, the article becomes difficult to read. Therefore,

  1. Please check the entire article for English. There are many mistakes and some phrases must be reformulated.
  2. The abstract must be reformulated to emphasize the novelty of the approach and the main finding, without repeating information.
  3. Introduction must be improved. It does not present enough information on the techniques used by other scientists for similar problems. Also, the novelty is not emphasized.
  4. Sections 2 and 3 must be put together under Research Methodology.
  5. Dissimilarity and entropy definitions should be introduced.
  6. Please provide information on the software used for modeling.

Author Response

We do appreciate your help on this paper. Your advice renders us many suggestive perspective views to make this paper present perfectly.

Reviewer 2 Report

The submission aimed at improving the detection of paddy rice by implementing a two-step classification approach where residual mislabelling at the first stage (direct classification) are resolved at a second stage (hybrid classification) using dynamic time warping. The study utilized multi-temporal multi-sensor (optical+SAR) data and employed three well-known machine learning algorithms at the two classification stages. Detection accuracies improved at the second stage (using DTW) compared to the first. My major comments/concerns are below:

  1. The submission suffers greatly from clarity. First, the reason/justification to employ the two stage classification, and use the DTW at the second stage is not well articulated. If the DTW indeed improves detection accuracy, I would expect authors to just go ahead and use it directly instead of going through an earlier classification. Second, the description of the DTW approach, and how it improves detection accuracy or resolves residual misclassification’s (at stage 1) is poor and confusing and unconvincing. It’s not clear how computation of the similarity between two time series can perform better that the use of the time series themselves. The use of multiple terminologies – e.g. multi-scale timing feature similarity, multi-scale time series feature similarity indicator, feature similarity index – adds to the confusion. I didn’t quite understand what ancillary
  2. The accuracy improvements observed after employing DTW is not worth the effort, I think (lines 491 to 500). Maximum increment of 3.1% was observed between the direct classification (without DTW) and the hybrid classification (with DTW). The difference for DT, for example, is just about 1%. I honestly don’t think that such savings warrant all the efforts and computational resources at stage 2
  3. Perhaps my biggest worry is that lack of discussion section where the results are discussed in light of other studies that employed a DTW. Apart from the mention of one paper that used the DTW (lines 283 – 286), the submission is virtually silent on how DTW fared in other papers and possible reasons for its adoption here.
  4. Whereas I appreciate that English may not be the first language of the speaker, I think tremendous improvement is required to improve the language; it might have added to my confusion.

Author Response

(The authors gave the same response as above.)

Reviewer 3 Report

Multi-temporal data fusion in MS and SAR images using Dy- namic Time Warping method for paddy rice classification

This manuscript uses multi-temporal MS and SAR imagery for paddy rice classification. Furthermore, using the Dynamic Time Wrapping (DTW), the authors have developed a „"multi-scale time series feature similarity index“. The results of this research display a feasible way to integrate multi-temporal radar with optical data for classification of paddy rice fields.

Review summary

While the topic of this manuscript is in principal interesting for the readers, currently written manuscript does not provide significant scientific contributions. Furthermore, Section Results and Discussion must be improved with more explanations and discussions.

  • Introduction: this part has a lot of enumerated technology (e.g., machine learning methods, remote sensing imagery, object-based classification, active and passive sensors) without any explanation in depth which is connected to a current research (e.g., usage of the aforementioned technology / data for paddy rice classification). Furthermore, the abbreviation 'etc' is overly used.
  • Section 2.1.2: after reading this Section, the readers do not know how many bands SPOT6 has, what is a spatial resolution. The reason for using these imagery in the research is because Remote Sensing Centre completed the processing of the imagery? („They are completed by the Central University Space Remote Sensing Center. Hence, this study decided to use them.“). Also, we do not know the product type of Sentinel-1 imagery
  • LN 163 – 166: optical/radar/texture features are enumerated, but we do not know which one
  • LN 175 and 178: the Authors mention twice „Four texture features (GLCM) are also used.“. The manuscript needs to be double checked from the Authors
  • Figure 5: In Part 1 feature selection is done, but is not mentioned in manuscript, Part 2, is Direct / Hybrid classification method connected only to NN?
  • LN 239: I think that this part: „Explain the classification methods (SVM, NN, and DT) in detail.“ is an instruction between the Authors. Once again, the manuscript needs to be double checked from the Authors
  • Section 3.5. must be one of the most important part of the manuscript because is provides information how the accuracy validation has been done. In a current form, we only know that confusion matrix and kappa value has been used
  • Figure 6: if we take a look on this Figure without looking at the Figure caption, we do not know what are the units on the y-axis, which polarization bands were used, etc.
  • LN 327: „classify the paddy rice in image classification. This study also proved this progress. Besides the CMFI and GESAVI, most of the ancillary indicators can provide useful information to detect paddy rice vs. non-paddy rice.“ The authors did not provide any quantitative information to confirm this statement
  • LN 374: using + sign in a manuscript?
  • Table 3: isn't it more convenient to use User's, Producer's accuracy as a complementary measures of omission / commission. Overall, I am deeply concerned about data imbalance in this research
  • LN 449: „From the results, it can compare the three classification methods, the best classification result is SVM. The accuracy and Kappa values are 94.43% and 0.80, respectively.“ However, Table 7 shows that DT obtained OA of 94.71% and kappa 0.81?
  • Section 4 is named Results and Discussion. However, in a current form, Discussion (e.g., comparison with similar research) is not present in a manuscript
  • Conclusions: this part is too long and the Authors repeat all the information presented in the manuscript and the readers do not receive the most important conclusions of the research. Why the Authors mention "multi-scale time series feature similarity index" in quotation marks, it looks like they are not convinced what was the scientific contribution of the manuscript.
  • LN 501: „The DTW index developed in this research can virtually provide an effective assistant for paddy field classification“ Can the Authors explain what they meant with the statement that DTW can virtually provide, what is the other way?
  • LN 506: „Unfortunately, the radar image data has a flaw in detecting vegetation areas.“ This part does not belong to the Conclusion, eventually to the Introduction with some research which investigated that problem.

Overall 17 references were used, and research about DTW is dated from 2011 [10] and 2016 [11], which speaks to itself about novelty in this manuscript. Furthermore, high accuracy results were obtained due to the data imbalance, which needs to be adjusted.

Author Response

(The authors gave the same response as above.)

Round 2

Reviewer 1 Report

Dear authors,

Thank you for providing the answers to my questions. Still, I have to point out the following.

  1. English must be polished by a native English speaker. There are issues like The Kappa is must be considered ...
  2. The abstract is incomprehensible. Please avoid putting many propositions in the same phrase.
  3. line 34-35. It is hard to conduct a large-scale survey in a short time - must be removed
  4. Repeating the same word in a phrase is boring. For example, the use... - lines 37-38. The phrase should be broken into two parts. There is no relationship between the two techniques. Machine learning is employed for modeling, not for monitoring. They may be employed separately. 
  5. line 39-40 - please remove it put in a single phrase what you want to communicate about machine learning.
  6. 5. Introduction does not show the research stage in the field of this specific research. I cannot find what other scientists did.
  7. Lines 395-397. Actually, we are interested in the best approach. Why perform a study if what we get is not the best result. So, please provide the answer to this question related to your assertion from lines 395-397. 
  8. the crop is a long-term characteristic - What do you mean? Crop is a land use.

I am sorry to see that the article should still nee to be checked and there are terminology issues.

Please revise this article from this viewpoint.

Reviewer 3 Report

Multi-temporal data fusion in MS and SAR images using Dynamic Time Warping method for paddy rice classification

This revised manuscript uses multi-temporal MS and SAR imagery for paddy rice classification. Furthermore, using the Dynamic Time Wrapping (DTW), the authors have developed a „"multi-scale time series feature similarity index“. The results of this research display a feasible way to integrate multi-temporal radar with optical data for classification of paddy rice fields.

Review summary

Still, this revised manuscript does not provide major scientific contributions. Hereinafter, I will mention some main weakness of the manuscript:

  • Figure 6: (a) – (b) figures look almost identical, and they should show VV and VH polarization for paddy rice / non paddy rice fields. Looking at this Figure, it can be sensed that the Results will be biased
  • Section 4.2.1 : Again, the Authors provide wrong information about direct classification method. In the body of the manuscript (LN 387) DT is mentioned that it performed the best, and results of the SVM are showed (OA 91.74 and Kappa of 0.72).
  • Table 3-5: in mentioned Tables a significant error is made from the Authors. Comparing v01 and v02, values remained the same regarding the commission/omission accuracy and the Authors just replaced it with User's/Producer's accuracy. This shows serious lack of knowledge about result interpretation since „User’s and producer’s accuracies provide the critically important information on class-specific accuracy, and these accuracy parameters are the complements of commission and omission error probabilities, respectively“ (Stehman 2009). Therefore, as an e.g., if in v01 commission error was 0.65 (Table 3 SVM method for paddy rice), than in v02 User's accuracy should be 0.35, etc.
  • The Authors made the change in Section naming from v01 to v02 – they left solely the Results section, and last Section is Summary and Conclusion. But again, the most important part is missing – it is Discussion

Overall, the Authors did accept the comments from the v01, and some modifications were made in the v02. But altogether, it is not incorporated well, e.g., the Authors did mentioned the SPOT bands (LN 135) but they were only introduced as abbreviations, Section 3.5 is very poorly described, etc.